# Liposomes for Tumor Targeted Therapy: A Review

**DOI:** 10.3390/ijms24032643

**Published:** 2023-01-31

**Authors:** Shile Wang, Yanyu Chen, Jiancheng Guo, Qinqin Huang

**Affiliations:** 1The Research and Application Center of Precision Medicine, The Second Affiliated Hospital of Zhengzhou University, Zhengzhou University, Jingba Road No. 2, Zhengzhou 450014, China; 2Precision Medicine Center, Academy of Medical Science, Zhengzhou University, Daxuebei Road No. 40, Zhengzhou 450052, China

**Keywords:** liposomes, drug delivery systems, cancer therapy

## Abstract

Liposomes, the most widely studied nano-drug carriers in drug delivery, are sphere-shaped vesicles consisting of one or more phospholipid bilayers. Compared with traditional drug delivery systems, liposomes exhibit prominent properties that include targeted delivery, high biocompatibility, biodegradability, easy functionalization, low toxicity, improvements in the sustained release of the drug it carries and improved therapeutic indices. In the wake of the rapid development of nanotechnology, the studies of liposome composition have become increasingly extensive. The molecular diversity of liposome composition, which includes long-circulating PEGylated liposomes, ligand-functionalized liposomes, stimuli-responsive liposomes, and advanced cell membrane-coated biomimetic nanocarriers, endows their drug delivery with unique physiological functions. This review describes the composition, types and preparation methods of liposomes, and discusses their targeting strategies in cancer therapy.

## 1. Introduction

Cancer is the second-most prominent cause of death in the world. According to the latest global cancer statistics from the GLOBLOCAN project in 2020, there are approximately 19.3 million new cases and 10 million cancer deaths worldwide [1]. In recent years, global cancer incidence and mortality rates have risen rapidly and begun to trend increasingly younger due to population growth and aging. At present, the main methods of cancer treatment include surgery, radiotherapy, chemotherapy and immunotherapy. However, these methods have shortcomings and challenges that include the difficult invasiveness of solid tumors, poor drug solubility, short circulation of chemotherapeutics drugs, multidrug resistance, nonspecific targeting, and systemic and local off-target side effects [2]. Therefore, there is an urgent need for innovative therapeutic strategies that can precisely localize the cancer site without damaging organs and tissues outside of a lesion.

With the rapid development of nanotechnology, the research of nanomaterials and nanomedical applications has recently become a scientific hot spot. Among nanomaterials, liposomes were the earliest nanodrug carriers to be discovered and are the most explored and widely functional to be used in targeted drug delivery systems. Liposomes were first discovered in the 1960s by Bengham et al., and later became the most widely used nanomedicine carriers in targeted drug delivery systems [3]. Liposomes are spherical vesicles composed of phospholipid bilayers with hydrophilic and hydrophobic characteristics [4]. Due to their amphiphilic feature, lipophilic drugs can be entrapped in phospholipid bilayers or adsorb on the liposome surface, whereas hydrophilic drugs can be encapsulated by the aqueous interior of vesicles [5]. Additionally, liposomes have good biocompatibility and biodegradability due to their phospholipid bilayer structure. Liposomes have advanced significantly in recent years as carriers of small molecule drugs, peptides, proteins [6], nucleic acids [7], etc. As drug vehicles, liposomes show prominent properties such as targeted delivery, high biocompatibility, biodegradability, easy functionalization, low toxicity and immunogenicity, all of which greatly increase the sustained release and therapeutic index of drugs [8,9]. However, liposomes are always cleared from the blood quickly. Therefore, to improve their stability in vivo, a series of studies have been carried out on the special modification of liposomes’ surfaces [10]. For instance, grafting the hydrophilic polymer polyethylene glycol (PEG) on the surface of liposomes can prevent the adsorption there of opsonins, thereby reducing mononuclear phagocyte system (MPS) uptake and achieving long-time circulation [11].

Currently, common functionalized liposomes include long-circulating PEGylated liposomes, ligand functionalized liposomes, stimuli responsive liposomes for highly effective targeted therapy, and advanced cell membrane-coated biomimetic nanocarriers (Figure 1). Meanwhile, the safety of nanoliposomes is also affected by several characteristics, including their size, composition, surface charge, stability, integration into tissues, and interactions with cells.

Antitumor drugs generally have serious side effects, thus their clinical use has been somewhat restricted. Due to their exceptional properties, liposomes can be used as a splendid drug carrier in the field of tumor therapy and have been widely used and studied in tumors. In recent years, with the rapid development of liposome technology, many liposome-based pharmaceutical preparations have been successfully transformed into clinical applications, and many liposome products are undergoing clinical trials [14]. The first liposome drug, Doxil^®^, was launched in the US in 1995 to treat patients with ovarian cancer and AIDS-associated Kaposi’s sarcoma [15]. DaunoXome^®^ was then developed by NeXstar Pharmaceuticals to deliver daunomycin. Later, more and more liposome products have been developed and applied in clinical practice, such as Mepact^®^, Marqibo^®^, Depocyt^®^ and Vyxeos^®^.

In addition to these mature clinical products and products undergoing clinical trials, some liposome preparations are being designed and developed. Girgis et al. [16] first used epidermal growth factor receptor (EGFR)-targeted photoactivated multi-inhibitor liposomes (TPMILs) to remediate desmoplasia in a pancreatic ductal adenocarcinoma model containing patient-derived pancreatic cancer-associated fibroblasts. TPMIL included irinotecan (IRI) and lipidated phenylporphyrin derivative photosensitizer. A series of experiments have demonstrated that a single EGFR–TPMIL construct is capable of remediating desmoplasia, controlling tumor development, and doubling survival [16]. Zhao et al. [17] coupled anti-programmed death protein 1 (PD-1) inhibitor and doxorubicin (DOX)-loaded liposomes to cell corpses, thereby developing “walking dead” triple-negative breast cancer cells. The delivery system significantly enhanced anti-tumor and anti-metastasis effects in vivo when employed for chemoimmunotherapy of metastatic breast cancer. The present review will briefly describe the composition and types of liposomes, the methods of preparation and the clinical applications.

## 2. Composition and Classification of Liposomes

Liposomes are mainly composed of phospholipids (glycerophospholipids and sphingomyelins). Phospholipids are amphiphilic molecules with hydrophobic tail groups and hydrophilic head groups [4]. These amphiphilic lipids spontaneously organize into liposomes in an aqueous environment, driven by hydrophobic interactions and other intermolecular interactions. These phospholipids can be synthesized entirely or partially from natural sources. The chemical properties of the phospholipids affect the biodistribution, clearance, drug release, permeability, and surface charge of liposomal formulations [9,11]. Meanwhile, the entrapment efficiency, toxicity and stability of liposomes are also affected by the type of phospholipids utilized in their preparation [9]. Natural-based liposomes, fabricated from unsaturated phosphatidylcholine species, provide bilayer structures with high permeability and low stability. However, liposomes made from saturated phospholipids, such as dipalmitoyl phosphatidylcholine, result in stiff and almost impenetrable bilayer structures [18,19,20].

In addition to phospholipids, sterols have been added to liposomes to modulate membrane fluidity and improve liposome stability [18]. Cholesterol is the most commonly used sterol in the formulation of liposomes. It promotes phospholipid packaging and bilayer formation, decreases membrane fluidity, reduces transmembrane transport of water-soluble drugs, and improves bilayer stability in biological fluids such as blood and plasma [9,15,18]. Meanwhile, cholesterol can also lessen the interaction of liposomes with proteins in vivo, reduce the phospholipid depletion of high-density lipoproteins, and inhibit macrophage digestion, thereby improving the stability of liposomes [18]. Furthermore, oligosaccharides, chitosan, and whey protein can be employed as liposome membranes to increase stability and regulate drug release.

Liposomes are vesicles that range in size from 20 nm to 2.5 μm. They may consist of one or several concentric or non-concentric membranes. The size of the vesicle and number of bilayers are acute factors that affect drug encapsulation effectiveness, clearance, and half-life in circulation [4]. Depending on the size and number of bilayers, liposomes can be classified into unilamellar vesicles, multilamellar vesicles (MLV, >500 nm), and multivesicular vesicles (MVV, >1000 nm). Unilamellar vesicles can be divided into small unilamellar vesicles (SUV, 20–100 nm), large unilamellar vesicles (LUV, 100–1000 nm), and giant unilamellar vesicles (GUV, >1000 nm) [9,21]. To work efficiently, liposomes need to be prepared at a certain size to allow absorption into cells [21]. SUV are the most commonly employed drug delivery vehicles because they have consistent drug encapsulation and release kinetics and extended circulation times [22].

## 3. Liposomes Preparation

### 3.1. Traditional Preparation Methods

There are a wide variety of traditional methods used to prepare liposomes, including thin-film hydration, reverse-phase evaporation, ethanol injection, sonication, extrusion, high-pressure homogenization, freeze-thaw, of which ultrasonic, extrusion, high-pressure homogenization and freeze-thaw can be prepared in combination with other methods (Table 1).

### 3.2. Microfluidic-Assisted Formation of Liposomes

The aforementioned traditional liposome production techniques often call for a number of postprocessing steps (such as extrusion, freeze-thaw, sonication, and/or high-pressure homogenization) to lessen the final liposome size and polydispersity [30]. Conventional preparation methods have many limitations, including cumbersome steps, inconsistent nanoparticle structure, poor reproducibility between batches, low drug encapsulation efficiency, and polydispersed size distribution. These issues have recently been resolved by other brand-new techniques for producing liposomes [31]. Microfluidics is a versatile technology for processing or manipulating microfluids in channel structures with dimensions of tens to hundreds of micrometers [32]. Fluids controlled by microfluidic technology have unique properties, such as laminar flow and droplets. Using these unique fluidic phenomena, we can finely control the mixing conditions of each solvent, precisely control the assembly of nanostructures and command batch quality to achieve uniformity in the size distribution of the synthesized particles. By adjusting the flow parameters in hydrodynamic focusing systems, such as total flow rate, flow rate ratio of mixed solutions of the solutions mixed, lipid solution concentration, and feature length in microfluidic channels, it is simple to generate nanoscale liposomes with smaller and narrower size distributions [31].

#### 3.2.1. Microfluidic Hydrodynamic Focusing (MHF)

The earliest examples of microfluidic technologies include hydration, pulsed jets, droplet emulsion transfer, and hydrodynamic focusing; however, these techniques frequently produce relatively large vesicular systems or microtubules, which are ineffective for nanodrug delivery [33]. Jahn et al. [34] first reported the fine control of liposomes by MHF. The two side channels containing aqueous solutions were vertically connected to the central channel with the lipids dissolved in alcohol. As the alcohol was mixed and diluted to a critical concentration by the aqueous solutions, the lipids spontaneously self-assembled into liposomes (Figure 2A) [21]. The size of the generated liposomes could be easily controlled to the range of 100–300 nm by adjusting the flow rate in the microfluidic channel [34]. For further investigation, Jahn et al. [35] modified their microfluidic system, which considerably increased its capacity to regulate liposome size and elucidated the mechanism by which to control liposome size and uniformity. They discovered that, as opposed to shear forces at the solvent–buffer interface, the generation of liposomes was dependent on the width of the focused alcohol stream and its diffusion mixing with the aqueous phase flow [35]. The 2D hydrodynamic flow-focusing stream can be produced using a classic chip-type device. This restricts control over liposome particle size distribution [36].

Some compounds may diffuse into their surroundings prior to the production of vesicles as the development of vesicles depends on the process of diffusion from the organic layer to the aqueous layer [37]. In addition, liposomes formed by this method may retain alcohols and cause stability problems [36]. To further improve productivity, Hood et al. [38] designed a three-dimensional microfluidic hydrodynamic focusing (3D-MHF) device, which was a concentric capillary array composed of seven identical borosilicate glass capillaries (Figure 2B). The liposome polydispersity produced by the 3D-MHF device was quadrupled, and the productivity was four orders of magnitude higher than the previous MHF method, due to the complete radial symmetry diffusion of alcohol solvated lipids into the aqueous stream [38]. Based on MHF, Koh et al. [39] fabricated a five-inlet polymeric microfluidic hydrodynamic focusing system to prepare oligonucleotides-containing liposomes with smaller size, narrower size distribution and more uniform structure.

**Figure 2 ijms-24-02643-f002:**
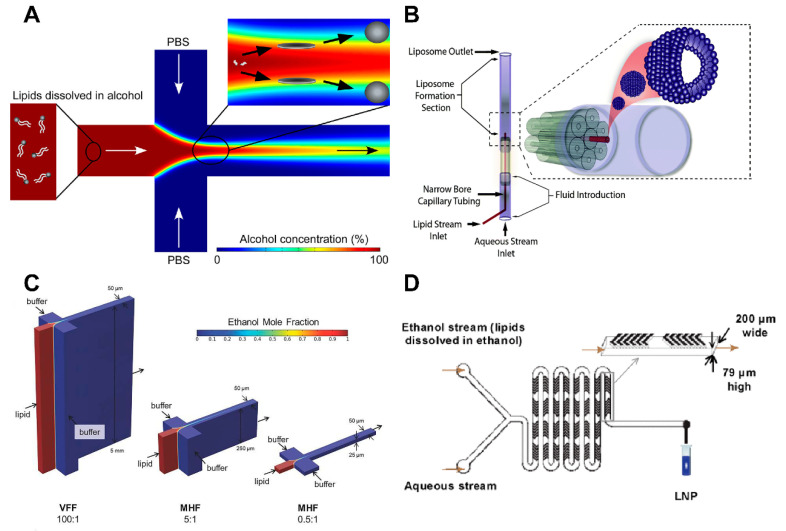
Schematic diagram of microfluidic methods. (**A**) Schematic of an example of the MHF method. Reproduced with permission from [40]. Copyright 2013, American Chemical Society. (**B**) An example of 3D-MHF. Reproduced with permission from [38]. Copyright 2014, Royal Society of Chemistry. (**C**) Vertical flow-focusing technique. Reproduced with permission from [30]. Copyright 2015, Wiley-VCH. (**D**) Schematic of lipid nanoparticles formulation process employing the staggered herringbone mixer (SHM) device. Reproduced with permission from [41]. Copyright 2012, American Chemical Society.

#### 3.2.2. Microfluidic Vertical Flow Focusing

The benefits of using microfluidic technology to prepare liposomes have been demonstrated by the creation of liposomes made of diverse lipids, however, commercialization of this technology has always been difficult due to its small production scale and limited throughput. On the basis of MHF, Hood et al. [30] designed a microfluidic vertical flow focusing device with a high aspect ratio (depth: width = 100:1) (Figure 2C). In this instance, the vertical flow focusing method was used to produce populations of small, unilamellar, and almost monodispersed liposomal nanoparticles at remarkably high production rates. The microfluidic vertical flow-focusing device successfully realized the concept of multi-layer vertical flow focusing by thermoplastic micromachining process to form high aspect ratio channels [30]. By adjusting the fluid flow ratio (alcohol–water volume flow velocity ratio) in the microfluidic network, liposomes with a modal diameter of 80–200 nm could be produced at rates as high as 1.6 mg/min^−1^ [30].

#### 3.2.3. Micromixer

After years of research, it has been found that liposomes produced by devices based on hydrodynamic flow focusing show great limitations in size and throughput. The microfluidic SHM designed by Stroock et al. [42] is a powerful and general approach to generate liposomes with smaller sizes, and the method can solve the above problems well. Zhigaltsev et al. [41] pioneered SHM as a scalable manufacturing method for the production of lipids of limited size (Figure 2D). The mixer uses chaotic advection. The chaotic advection causes the water flow to stretch and fold over the channel cross-section, effectively increasing mass transfer while increasing the herringbone structure at the bottom of the channel [37]. Maeki et al. found that the size and polydispersity index of liposomes prepared by SHM decreased simultaneously with the increase of flow rate ratio [21,43]. However, there are practical limitations in terms of cost and production speed of such micromixers [44]. To solve these problems, a toroidal mixer can be used as an alternative design. Generating more vortices and centrifugal forces between the inner columns of the cartridge induces chaotic advection, which improves mixing and higher throughput [44].

## 4. Functionalized Liposomes

To enhance the clinical applications of liposomes, researchers have functionalized liposomes in specific ways. The common types include long-circulating liposomes, actively targeted liposomes, stimuli-responsive liposomes, and cell membrane-coated liposomes.

### 4.1. Long-Circulating Liposomes

The neovasculature formed by rapid tumor growth contains wide fenestrations and restricted lymphatic drainage, allowing nanodrug carriers to flow through the comparatively leaky blood arteries of the tumor and consolidate at the target site [45,46]. This phenomenon is known as the enhanced permeability and retention (EPR) effect. Indeed, tumor absorption of liposomes is essentially dependent on the EPR effect. However, when administered intravenously, liposomes are quickly removed by MPS in the liver and spleen [47]. After entering the circulation, MPS uptakes liposomes by recognizing plasma proteins and complement components bound to a liposome’s surface [11,48,49].

Based on this principle, it was found that liposome surface conjugation with hydrophilic polymers or a glycolipid could decrease MPS uptake of liposomes, thus allowing their long-term circulation in the blood. These hydrophilic polymers or glycolipids can occupy the space on the liposome surface, thereby excluding other macromolecules from this space [11]. Consequently, the binding of liposomes to plasma proteins is hindered, thus prolonging their circulation life. Huayluorinic acid, polyvinyl alcohol, and PEG are thought to be the best models for liposome steric protection [9]. Among these, PEGylation is a widely used method to achieve long circulation inside liposomes. PEG is a hydrophilic polymer that reduces the adhesion of nanoparticles to mucin fibers in the mucus mesh and increases mucus penetration, allowing particles to quickly penetrate mucus [50,51]. Thus, PEGylated liposomes enable sustained and targeted mucosal drug delivery. In order to improve the efficiency of mucosal administration, and in addition to promoting the mucus permeability of the particles, these liposomes can also improve the mucosal adhesion and prolong the residence time of the preparation in the mucosa [52]. In recent years, it has been found that polymers functionalized with maleimide can form covalent linkages with thiol groups present in mucins to exhibit excellent mucosal adhesion properties [53,54]. Kaldybekov et al. [52] showed good performance in bladder mucosal administration with maleimide-functionalized PEGylated liposomes.

In addition, PEG functionalization on the liposome surface can alter the pharmacokinetics of liposomes, resulting in slower release rates. Although PEG-functionalized liposomes show many advantages, deepening of research has shown that there are many problems with PEGylation, including the slow release of loaded drugs, the failure of fusion with endosomes after internalization [55], and the steric hindrance of PEG chains inhibiting the uptake of liposomes by target cells [56].

Liposomes prepared from gangliosides provide a layer of surface-charged groups with longer blood-circulation time. In addition, Oku and Namba have found that palmityl-D-glucuronide-modified liposomes decreased the MPS recognition rate and reduced the liver accumulation in vivo, thus increasing tumor accumulation, which in turn enhances the therapeutic efficacy of DOX in long-circulating liposomes [11,57]. Nowadays, chitosan coating the surface of negatively charged liposomes has been found to improve the stability of liposomes and ensure the sustained release of drugs [58]. Since chitosan is positively charged, it can reduce liposomes uptake by cells. Moreover, chitosan mucoadhesive properties will prolong the residence times at the site of drug absorption due to the increased contact of the carrier with the absorbing site [58]. Based on these unique biological characteristics, chitosan-coated, drug-loaded liposomes have opened novel avenues for ocular drug delivery [59]. In addition, the albumin coating of liposomes reduces the binding of serum proteins to liposomes and facilitates an increase in circulation time in the blood. Similarly, liposomes coated with the combination of polyvinyl alcohol and PEG also show a better long-circulating function. Giulimondi et al. [60] exposed DNA-decorated lipoplexes to human plasma. These complexes were covered with an opsonin-deficient protein corona and the final synthesis product was a type of lipoplex covered by a proteonucleotidic corona, or “proteoDNAsome”. These experiments proved that this lipoplex has a more pronounced ability to evade capture by immune cells in vivo than PEGylated lipoplexes.

### 4.2. Actively Targeted Liposomes

Many liposome nanocarriers have been approved to deliver antitumor drugs in a passively targeted manner. However, passive targeting does not distinguish between normal and diseased cells. Therefore, targeted liposomes have been developed to increase the localization and accumulation of liposome drugs in tumor tissues [9,61]. Active targeting liposomes can improve drug efficacy and reduce side effects through two strategies: by targeting the overexpressed surface receptors of cancer cells and by targeting the tumor microenvironment (TME) [62]. Ligands for actively targeted liposomes include antibodies, proteins, peptides, vitamins, growth factors, aptamers, etc.

#### 4.2.1. Surface Modification of Liposomes with Antibodies

Most studies in the field of targeted liposomes involve the use of multiple antibodies for tumor targeting purposes. Conjugation of anti-CD22 monoclonal antibodies to PEGylated DOX-loaded liposomes increases the accumulation of DOX in Non-Hodgkin’s Lymphoma xenografts and improves antitumor efficacy without increasing toxicity [63]. Human Epidermal Growth Factor Receptor 2 (HER2) is highly expressed in a significant proportion of breast, ovarian, and gastric cancers and is under-expressed in normal cells, so targeted drug delivery for cancer can be carried out by coupling ligands targeting HER2 [64]. Trastuzumab, an anti-HER2 antibody, is conjugated to paclitaxel-loaded PEGylated liposomes to increase drug accumulation and antitumor efficacy in tumors [65]. In addition, the combination of a single-chain fragment antibody against HER2 with DOX-loaded liposomes increases the accumulation of DOX in breast cancer tumors and has better tumor control than free DOX and untargeted liposomal DOX [66]. Milatuzumab, a CD74 antagonist monoclonal antibody, is highly toxic to chronic lymphocytic leukemia cells when added to liposomes. CD19 is expressed in most acute B-ALL cells, does not shed from the surface of malignant cells and is internalized after antibody binding, so CD19 can be used as a target antigen [67]. Vinu et al. [67] found that anti-CD19 antibodies conjugated with DOX-encapsulated liposomes rapidly internalized in a clathrin-dependent manner, which could lead to an increase in DOX levels in tumor cells.

#### 4.2.2. Surface Modification of Liposomes with Folic Acid

Folic acid is a water-soluble vitamin that binds to the folate receptor (FR) with high affinity and induces receptor-mediated endocytosis. The FR is rarely expressed in normal cells, but is overexpressed in certain lung, ovarian, colon, breast, brain and kidney tumors [68,69]. Studies have found that FR can be expressed in different cells and can be selectively targeted. For instance, folate receptor alpha is expressed in a number of epithelial cancers, particularly epithelial ovarian cancer, whereas folate receptor beta is mainly restricted to macrophages [68,70]. Folate-targeted liposomes have successfully delivered chemotherapeutic agents as well as genes, antisense oligonucleotides and radionuclides to tumor cells with overexpressed FR [71]. Wu et al. [72] compared FR-targeted paclitaxel liposomes with untargeted paclitaxel-loaded liposomes and found that the former had greater cytotoxicity activity and longer terminal half-life. Kareen et al. [73] demonstrated that folate-targeted liposomal DOX had the potential to enhance in vivo anticancer drug delivery in the treatment of KB tumor mouse models. Folic acid conjugated PEGylated liposomes exhibit high antitumor efficacy through their increased drug solubility and increased circulating half-life.

#### 4.2.3. Surface Modification of Liposomes with Transferrin (Tf)

Tf can specifically bind to the transferrin receptor (TfR), which is highly expressed on the surface of tumor cells. Thus, it can be bound to drug delivery systems for specific targeting [30]. TfRs are overexpressed in tumor and leukemia cells. Tf conjugation to DOX-loaded stealth liposomes increases DOX delivery to tumors and inhibits tumor growth compared with untargeted DOX-loaded liposomes [10,74]. Compared with untargeted liposomes, cisplatin-encapsulated transferrin-conjugated PEGylated liposomes not only increase accumulation in free tumor cells in ascites, but also in solid tumor tissues of the greater omentum [75]. In Colon 26 tumor-bearing animals, Tf-PEG-liposomes have shown a prolonged residence period in the circulation and low reticuloendothelial system absorption, resulting in improved extravasation of the liposomes into the solid tumor tissue and a boosting of the drug uptake into tumor cells through endocytosis [76]. TfRs are also expressed in some normal cells, so Tf liposomes may enhance intracellular DOX uptake and cytotoxicity to these normal cells. Particular attention should be paid to its cardiotoxicity.

#### 4.2.4. Surface Modification of Liposomes with Peptides

Peptide-mediated liposomes are widely used to deliver anticancer drugs and siRNA due to their various advantages. Their structure is modifiable and smaller than antibodies, making them easy to chemically synthesize in order to meet targeting needs. To achieve a more effective targeting of nanocarriers, phage display libraries have recently been studied to facilitate peptide-mediated liposomal drug delivery. Receptors for luteinizing hormone-releasing hormone (LHRH) are overexpressed in breast, ovarian, and prostate cancer cells, but are not expressed detectably in most visceral organs [77,78]. PEGylated paclitaxel-loaded liposomes conjugated with LHRH peptides significantly enhance tumor accumulation and the anticancer efficacy of all delivery systems and minimize adverse side effects on healthy tissues over untargeted paclitaxel-loaded liposomes [79]. Cell-penetrating peptides translocate a variety of cargoes into the cells in a non-invasive manner without the use of receptors to enhance carrier internalization [7]. Penetratin, a type of cell-penetrating peptide, can penetrate neurons and accumulate in the nucleus [80]. Bruna et al. [7] modified the surface of liposomes with Tf protein and Penetratin for the efficient delivery of targeted genes to neuronal cells. Bifunctional Penetratin Tf-liposomes can protect plasmid DNA from enzymatic degradation while providing low cytotoxicity and high transfection efficiency.

#### 4.2.5. Surface Modification of Liposomes with Aptamers

Aptamers are short single-stranded DNA or RNA that can specifically bind to corresponding targets such as proteins and cells through their unique three-dimensional pattern [62]. Aptamers have been demonstrated to be superior and the fastest novel tools for efficiently targeting cancer biomarkers and are employed as effective ligands for drug delivery and anticancer therapy [62]. Aptamers possess several intrinsic advantages, including easy synthesis, convenient modification, high programmability, and good biocompatibility [81]. Dihua et al. [82] developed a nucleic acid aptamer, sgc8, for the highly specific recognition of human acute lymphatic leukemia cells. To investigate the ability of the tumor-targeting IL-4Rα-aptamer liposome-cytosine phosphate guanine oligodeoxynucleotides (CpG ODN) delivery system to introduce CpG into tumors and overcome immunosuppressive TME, Liu et al. [83] found that IL-4Rα-liposome-CpG therapy showed enhanced antitumor activity in CT26 tumor-bearing mice compared with the control group.

#### 4.2.6. Targeting the TME

In addition to targeting cancer cell surface receptors, tumor-specific active targeting can also target the TME. Nanomedicine can be used to target TME-specific molecular markers such as extracellular matrix components, tumor-specific pathophysiological circumstances, and TME-specific enzymes [45]. Liposomes can block tumor diffusion by targeting the tumor vasculature [62]. Targeting TME with chemical ligands shows great potential in improving the targeting efficiency of drug delivery and the efficacy of cancer therapy due to the many unique characteristics of TME components (Table 2) [45].

### 4.3. Stimuli-Responsive Liposomes

Surface modification of liposomes can attach ligands for site-specific targeting. In addition, chemical moieties can be attached to the surface of the liposomes, which are called stimuli-responsive liposomes (smart liposomes), that will respond to various different stimuli [87]. When certain external or internal signals occur, stimuli-responsive liposomes release their loaded contents at specific sites. These stimuli can be internal (e.g., enzyme activity, pH changes, redox potential, etc.) or external (e.g., temperature, light, magnetic field, or ultrasound) [88]. Stimuli-responsive liposomes can trigger drug release and increase the controllability of treatment in space and time. The intelligently engineered liposomes are important for delivering chemotherapy drugs because they provide higher concentrations of diagnostic or therapeutic drugs at the site of disease. Over the past few years, many different stimuli-responsive liposomes have been developed (Figure 3) (Table 3).

### 4.4. Cell Membrane-Coated Liposomes

Nanotechnology has been increasingly used for drug delivery to further improve clinical efficacy. Surface modification of liposomes can prolong blood circulation time and enhance targeted drug delivery. Most nanoparticle-based delivery methods concentrate largely on synthetic techniques, which frequently need cumbersome chemical production [108]. In addition, synthetic materials are not satisfactorily biocompatible and are difficult to manufacture on a large scale. With the in-depth study of nanomedicine, it has been discovered that combining synthetic nanoparticles with natural biomaterials, such as cell membranes, directly imparts highly complex functional surfaces and effective biological interfaces with nanocarriers. The platform of cell membrane-camouflaged nanoparticles has emerged as an innovative delivery method with the potential to enhance therapeutic effectiveness for the treatment of a range of ailments. Cell membrane-coated liposomes are a new class of biomimetic nanoliposomes that combine the customizability and flexibility of synthetic materials, as well as the functionality and complexity of natural materials [109]. They offer long cycle times, high biocompatibility, targeting capabilities, and simultaneous multidrug delivery.

At present, many types of membranes have been used to construct biomimetic nanoliposomes for cancer treatment, including membranes for red blood cells (RBCs), leukocytes, platelets, cancer cells, bacteria, stem cells, etc. Liposomes have diverse functions depending on the kind of cell membrane.

#### 4.4.1. RBC Membrane-Coated Liposomes

In 2011, a cell membrane coating technology that used RBC membranes as membrane materials was first reported [110]. RBCs, the most abundant cells in the blood, are mainly responsible for oxygen delivery within the body and have a lifespan of up to 120 days in humans. This technology inhibits immune attack to prolong blood circulation time, mediated by markers including CD47 molecules on the surface of RBCs and a series of complementary regulatory proteins [110]. This property of the erythrocyte membrane is highly desired for nanodrug delivery systems. Cell membrane coating technology provides a top-down approach by transferring the entire erythrocyte membrane directly to the synthetic nanoparticles. Li et al. [111] anchored the peptide ligand derived from the cytoplasmic protein P4.2 on the surface of cationic liposomes. The peptide of P4.2 specifically recognized the cytoplasmic domain of erythrocyte key transmembrane receptor band 3, thereby preparing RBC membrane-coated liposomes (Figure 4A). RBC membrane-coated liposomes exhibit an appropriate size distribution and high stability, resulting in better circulation time compared with conventional PEGylated liposomes [111].

#### 4.4.2. Leukocyte Membrane-Coated Liposomes

Leukocytes, which include many diverse subsets including neutrophils, dendritic cells, macrophages, eosinophils, lymphocytes and others, are another major component of cells in the blood circulation. Leukocytes are widely found in blood vessels, lymphatic vessels and other tissues. The majority of leukocytes are capable of deformation movement, which facilitates their easy migration between lymphatic and blood vessels. Leukocytes make their home at the lesion sites, penetrate the vascular system, evade immune clearance, and target metastatic cancer cells through interaction of VCAM-1-α4 integrins [108]. Therefore, leukocyte membrane-coated liposomes are usually proposed for cancer drug delivery. VCAM-1 is highly expressed in murine 4T1 breast cancer cells, therefore, in vivo, macrophage membranes retaining integrin α4β1 enable liposomes to target metastatic cells and produce a notable inhibitory effect on lung metastasis of breast cancer [112] (Figure 4B). Due to their inflammatory chemotaxis, tumor-associated macrophages are a suitable drug delivery carrier for tumor targeting. For example, liposomes loaded with anticancer drug emtansine are coated with membranes from RAW264.7 mouse macrophages. NK cells can attack cancer cells directly through inhibitory and activating receptors on their cell surfaces or kill cells that have not been previously sensitized [113]. A biomimetic system of fusogenic liposomes loaded with DOX has demonstrated excellent tumor-homing efficacy, high affinity for malignancy, and anticancer activity when coated with NK cell membranes [113,114].

#### 4.4.3. Platelet Membrane-Coated Liposomes

Platelets are cell debris that are detached from the cytoplasm of mature megakaryocytes in the body’s bone marrow. In the blood circulation system, platelets mainly play a role in hemostasis and coagulation, supporting nutrition and repairing vascular endothelium. In addition, platelets also play an important role in regulating the body’s host immunity and inflammatory response [115]. Because the platelet membrane has a CD47 receptor, it has the ability to escape phagocytosis in the systemic circulation. In addition, the platelet membrane selectively adheres to the vasculatures of the disease sites, and can also specifically aggregate around circulating tumor cells through P-selection and CD44 receptors [108]. Guo et al. [116] have successfully prepared a biomimetic drug delivery system for cancer treatment using a platelet membrane as the outer shell and an alkyl radical generator-loaded C-TiO2 hollow nanoshell as the inner core. Yang et al. [117] prepared platelet membrane-coated hybrid microbubbles (Pla-MBs) by ultrasound-assisted recombination of liposomes and platelets. Pla-MBs were supplied with numerous adhesive receptors by coating with platelet membranes, allowing them to target and detect sepsis-induced acute kidney damage [117].

#### 4.4.4. Cancer Cell Membrane-Coated Liposomes

Cell coating technology is applicable to membranes of cancer cells as well as blood-derived cells. In recent years, scientists have also used cancer cell membranes as biomimetic membranes to wrap nanoparticles. Cancer cells have an inherent feature that allows them to cling to one another, causing tumors to develop indefinitely [114]. Cancer cells have multiple tumor-associated antigens on their surfaces, allowing them to target specific tumor locations via intrinsic homotypic binding. It is possible to deliver tumor antigens to the tumor site using liposomes coated with cancer cell membranes that have undergone in vitro processing. This will activate the innate immune response, which will destroy the cancer cells. Tang et al. [118] have developed a biomimetic mesoporous nano-enzyme system coated with cancer cells for self-enhancing catalytic therapy of drug-resistant tumors. For tumor cell-selective glycan engineering, Liu et al. [119] created pH-responsive azidosugar liposomes disguised with natural cancer-cell membrane. The biomimetic liposomes were able to detect lung metastases of malignancies in addition to selective glycan imaging of various cancer cells and breast cancer subtypes [119]. Zhang et al. [120] constructed an integrated hybrid nanovesicle cancer cell membrane liposome by fusing cancer cell membranes and a charge-reversal liposome membrane of matrix metalloproteinase 9 (MMP-9) switchable peptide (Figure 4C). This cancer liposome has a number of distinguishing and desired characteristics, including a long circulation half-life, effective homogenous lung cancer-targeting capacity, outstanding biocompatibility, high tumor accumulation, and more [120].

**Figure 4 ijms-24-02643-f004:**
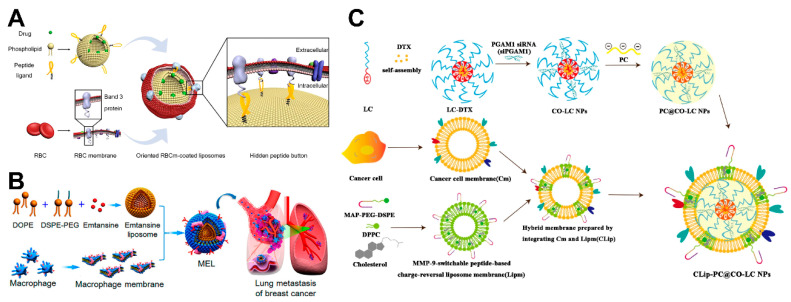
Schematic of several cell membrane-coated liposomes. (**A**) Schematic diagram of erythrocyte membrane and cationic liposome assembly. Reproduced with permission from [111]. Copyright 2019, American Chemical Society. (**B**) Schematic diagram of macrophage membrane-coated liposomes. Reproduced with permission from [112]. Copyright 2016, American Chemical Society. (**C**) Schematic diagram of membrane-coated liposomes in cancer cells. Reproduced with permission from [120]. Copyright 2021, the author(s).

#### 4.4.5. Stem Cell Membrane-Coated Liposomes

During multilineage differentiation, stem cells have a great capacity for self-replication. At various phases of development, stem cells have a natural active target impact on solid tumors [108]. Therefore, coating stem cell membranes on nanoparticles makes it possible to fabricate nanocarriers with similar targeting functions. Mesenchymal stem cells (MSCs) have the unique ability to home and engraft in tumor stroma [114]. As a result, MSCs are regarded as viable therapeutic vehicles for the targeting of major cells in a variety of pathological locales. Wu et al. [121] functionalized the curcumin-loaded liposomes with the membranes of MSCs, enabling biomimetic vesicles to have multiple functions such as long circulation, targeting and controlled drug release. Liu et al. [122] constructed a macrophage immunity reconstruction strategy by activating macrophages with stem cell biomimetic liposomes carrying levamisole for leukemia therapy. In this study, biomimetic liposomes garner unique properties such as long circulation, low immunogenicity and homing effect by incorporating the membranes of MSCs [122]. A new method, known as membrane fusion, has recently been proposed to provide targeted characteristics for liposomes by fusing them into cell membrane vesicles, resulting in liposomes [123]. Membrane fusion liposomes have the advantages of good stability, long circulation half-life and low immunogenicity. Liposomes have an exact replica of the targeting properties of membrane vesicles [124].

#### 4.4.6. Hybrid Cell Membrane-Coated Liposomes

Many of the cell types discussed so far highlight the unique functional properties of nanocarriers that can be given by cell membrane coating technology. Sometimes, it is also desirable to combine the functions of multiple cell types into one nanoparticle, that is, to integrate two different biofilms to encapsulate nanoparticles and maximize function. According to research, core coating or the integration of liposome nanocarriers has no influence on the distinct properties of various cell membranes. To increase solubility and targeting, a liposome-based delivery system was disguised by a hybrid cell membrane (RBC membrane and cancer cell membrane) and RGD surface modifications [125]. The hybrid membrane-coated liposomes have an extended half-life, increased immune evasion and targeted ability at the tumor site as well as good antitumor activity without side effects to normal tissue [125]. He et al. [126] fused together the membranes of leukocytes, cancer cells, and liposome nanocarriers simultaneously to prepare leukosomes. Compared with Liposomal nanoparticles made of exogenous phospholipids only or in combination with one type of cell membrane, the leukosomes prolonged blood circulation time, improved the efficiency of tumor site accumulation, and effectively inhibited tumor growth [126].

## 5. Liposomes in Clinical Applications

### 5.1. Liposomes for Cancer Therapy

Compared with traditional drug delivery systems, liposomes exhibit better biological properties, including high biocompatibility, low toxicity, easy surface modification, high targeting, encapsulation of various drugs, and protection from degradation. Given these merits, several liposomal drug products have been successfully approved and used in clinics over recent decades (Table 4). Among these, Doxil (Doxorubicin HCl liposome injection) was the first liposomal product approved by the FDA in 1995 [15]. In fact, the progress of nanoliposomes has promoted the treatment of cancer.

ThermoDox^®^, a liposomal drug carrier with lyso-thermosensitive liposomal DOX, is the first heat-activated formulation to be used in human clinical studies [127]. When the temperature rises to 40–45 °C, the thermosensitive liposomes rapidly change their structure to release DOX into the targeted site [128]. In clinical trials, ThermoDox^®^ has been combined with radiofrequency ablation to treat hepatocellular carcinoma. However, the OPTIMA trial did not achieve the endpoints, though this did not mean that ThermoDox^®^ was not feasible. Early clinical data of ThermoDox^®^ has demonstrated feasibility, safety and activity [127]. Currently, in a new phase I trial, the University of Oxford explored the safety and feasibility of combining ThermoDox^®^ with focused ultrasound for the treatment of non-resectable pancreatic cancer [129]. Josanne S et al. [130] first explored the phase I feasibility study of a combination of ThermoDox^®^, cyclophosphamide, and magnetic resonance-guided high-intensity focused ultrasound-induced hyperthermia in patients with stage IV breast cancer.

Myocet^®^, a non-PEGylated liposomal DOX, is combined with cyclophosphamide for first-line treatment of patients with metastatic breast cancer [131,132]. Compared with free DOX and cyclophosphamide, Myocet^®^ reduces cardiotoxicity and retains its antitumor efficacy [133]. EndoTAG-1™, a paclitaxel-loaded cationic liposome formulation, has been approved by the FDA. Cationic liposomes have been shown to target angiogenesis endothelial cells specifically, therefore, EndoTAG-1™ can inhibit tumor growth and metastasis by reducing tumor blood supply [134]. Sebastian et al. [134] demonstrated, in a phase I/II trial of EndoTAG-1, that infusion of 32 mg total lipid/kg body weight EndoTAG-1 was safe and had an antiangiogenic effect in human head and neck squamous cell carcinoma.

Cisplatin is a highly cytotoxic drug that inhibits DNA synthesis in tumor cells by creating inter-and intrachain cross-links. Therefore, cisplatin is still the first-line chemotherapy drug for various cancer types. However, cisplatin has serious side effects including nephrotoxicity, nausea and vomiting, ototoxicity, and allergic responses. To reduce the systemic toxicity of cisplatin, cisplatin liposomal preparations have been developed. Lipoplatinum is an FDA-approved cytotoxic agent that prolongs the circulating half-life of drugs, increases cell permeability, and accumulates drugs in tumor tissue [135]. Yang et al. [136] encapsulated the first-generation platinum anticancer agent cisplatin and phenethyl isothiocyanate in liposomes to treat non-small cell lung carcinomas. The liposomal preparation enhanced the toxicity of this doublet to NCI-H596 non-small cell lung carcinomas cells.

Onivyde™, an IRI liposome injection, was approved by the FDA in 2015 for the second-line treatment of pancreatic ductal adenocarcinoma. It is a long-circulating liposomal topoisomerase inhibitor that blocks DNA replication in cancer cells. Onivyde™ can increase the accumulation of IRI at the tumor site through the EPR effect. In human colon (HT29) and breast (BT474) cancer xenograft models, compared with free IRI, the liposomal IRI has an increased drug loading and prolonged drug half-life, resulting in a significant increase in cytotoxic activity [14]. Onivyde™, in combination with leucovorin and fluorouracil, is intended for the treatment of patients with metastatic adenocarcinoma of the pancreas who have progressed following gemcitabine-based therapy [14]. On 9 November 2022, French pharmaceutical company Ipsen announced that the Phase III clinical trial of Onivyde™ plus 5-fluorouracil/calcium leucovorin and oxaliplatin, as the first-line treatment for metastatic pancreatic ductal adenocarcinoma, had reached the primary endpoint.

In addition to clinical liposomal preparations, many novel liposomal drug delivery systems for cancer therapy are being developed in the laboratory. Platinum nanoparticles (nano-Pt) are highly cytotoxic and kill cancer cells by leaching Pt ions under low pH conditions [137]. Meanwhile, nano-Pt is also a catalase-like nanozyme [138]. Therefore, it can be used as an oxygen-replenishing nanomaterial to solve the problem of hypoxia limitation in photodynamic therapy. Liu et al. [138] prepared biomimetic liposome nanoplatinum by encapsulating nano-Pt and the photosensitizer vitepofen in macrophage membrane-coated liposomes. This liposome delivery system achieved deeper tumor tissue penetration while enhancing chemotherapy effectiveness with nano-Pt catalyzed oxygen delivery. Coupling MT1-MMP-activated cilengitide (MC) to DOX-loaded thermosensitive liposomes produces a novel smart nanovesicle MC-T-DOX, which can improve tumor blood perfusion, drug delivery, and treatment of pancreatic cancer by selectively stimulating tumor angiogenesis [139]. Shen et al. [140] proposed a PEGylated liposome loaded with mannose and levamisole hydrochloride to inhibit tumor growth. This suppresses tumor growth by restraining glycolysis and mitochondrial energy metabolism in cancer cells and macrophages. In addition, liposomes, used with radiotherapy, not only enhance the therapeutic effect of local tumors, but also increase the immune response to inhibit metastatic lesions [140]. Xing et al. [141] developed a liposome-based, light-triggered efficient sequential delivery method for multimodal chemotherapy, antiangiogenic and anti-myeloid-derived suppressor cell therapy in melanoma. This delivery strategy demonstrates the effectiveness of cancer multimodal therapy targeting multiple targets at different spatial locations in the TME [141].

### 5.2. Liposomes for Vaccines

Due to their adaptability and excellent biocompatibility, liposomes can be utilized as adjuvants in vaccines. It has been demonstrated that the encapsulation of peptide antigens or viral membrane proteins into liposomes results in humoral and cell-mediated immune responses, as well as strong and long-lasting immunity against the pathogen [142]. DNA/RNA and protein payloads may be shielded against biodegradation by liposome formulations. Furthermore, their transfection effectiveness may be improved by altering surface charge, size, and lipid structure [9]. Liposomes were initially described as immunological adjuvants by Gregoriadis and Allison in 1974 [143]. With the continuous development of nanotechnology in the medical field, new liposomes have been developed. At least eight liposome-based adjuvant systems are now authorized for use in humans or being tested in clinical settings.

In the midst of the SARS-CoV-2 viral pandemic, the first vaccines based on lipid nanoparticles and mRNA of the virus spike’s full-size S-protein were promptly put into the clinic [144]. Due to their extreme fragility, mRNA molecules require the assistance of a nano-delivery vector to be properly transferred to the target region and perform. Liposomes are used to protect the COVID-19 mRNA-based vaccines, increasing their in vitro and in vivo stability.

## 6. Challenges for Liposomes

With the deepening of nanotechnology research, liposomes as anti-tumor drug carrier treatment approaches and methods are increasingly widely used and have formed a phased development process. However, few lipid preparation agents have been successfully marketed, indicating that the development of liposomes still faces difficulties and challenges in the process of transformation from scientific research to clinical application. Among the many challenges, the stability of liposomes is crucial, which determines the changes in drug load, permeability, and drug release rate during the preparation, storage and metabolism of drug-loaded liposomes. The physical and chemical instability of liposomes greatly limits their clinical application as drug carriers (Table 5). The liposome membrane is a dynamic phospholipid membrane. Phospholipids can continuously undergo transmembrane movement, thereby inducing liposome particles to aggregate, sediment and form unstable phospholipid membranes, resulting in the instability of the physical morphological structure of liposomes. Oxidation and hydrolysis of phospholipids are the two main mechanisms that lead to the chemical instability of liposomes [145]. In addition, selecting the appropriate method for sterilization of liposome preparations is a huge challenge. At present, sterilization methods include steam sterilization, filtration sterilization and γ-ray radiation sterilization. Liposomes are susceptible to a variety of chemical and physical degradation mechanisms. The use of heat, radiation and toxic chemicals may alter the physicochemical properties of liposomal components and lead to increased toxicity of the final product.

## 7. Conclusions

Understanding the exact mechanism by which liposomes reach tumor sites and release loaded drugs is critical to resolving existing cancer therapy problems. Today, several liposome delivery technologies targeted for tumor targeting are being tested in clinical trials. With in-depth study, functionalized liposomes have evolved from simple vesicle structures to stimuli-responsive and cell membrane-coated liposomes. We can produce multifunctional nanocarriers by coupling numerous types of ligands on a single carrier using various surface modification methods of nanomaterials. In tumor treatment, multifunctional liposomes with prolonged release, targeted distribution, triggered release, and synergistic actions employing diverse surface functionalization and modification approaches will be essential. Liposomes have undergone multiple evolutions in terms of ingredients and manufacturing technique to overcome their early limitations. Nevertheless, the critical challenges of liposomes are their physical and chemical stability. Consequently, there is an essential requirement to develop liposomes with high stability, which has a significant impact on their clinical applications.

## Figures and Tables

**Figure 1 ijms-24-02643-f001:**
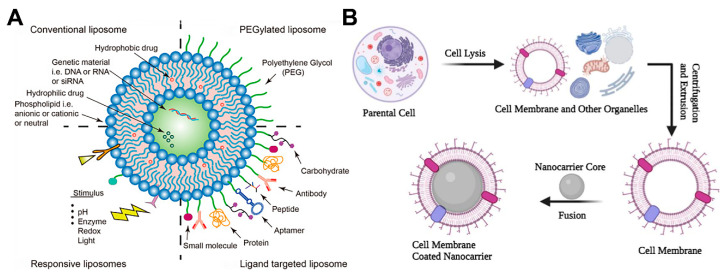
Schematic diagram of functionalized liposomes. (**A**) Strategies of multifunctional liposome delivery for cancer therapy. Reproduced with permission from reference [12]. Copyright 2020, the author(s). (**B**) Cell membrane-coated nanoparticles. Reproduced with permission from [13]. Copyright 2022, the author(s).

**Figure 3 ijms-24-02643-f003:**
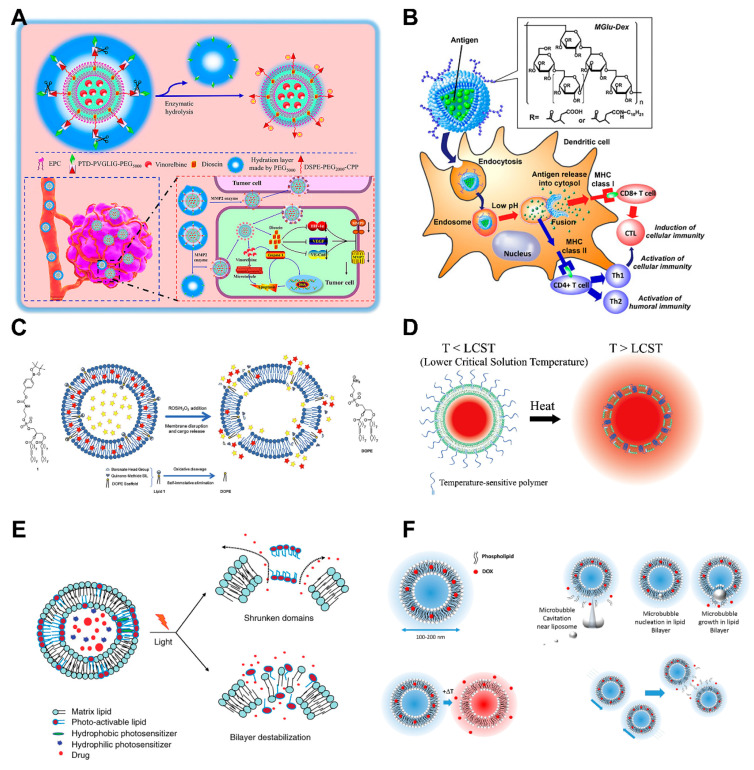
Demonstration diagrams of various stimulus-responsive liposomes. (**A**) Liposomes for enzyme-triggered drug delivery. Reproduced with permission from [89]. Copyright 2020, Dove Medical Press Limited. (**B**) Liposomes for PH-triggered drug delivery. Reproduced with permission from [90]. Copyright 2013, Elsevier B.V. (**C**) Reactive oxygen species (ROS)-responsive liposomes. Reproduced with permission from [91]. Copyright 2020, American Chemical Society. (**D**) Illustration of drug release from polymer-modified thermosensitive liposomes. Reproduced with permission from [92]. Copyright 2013, Elsevier B.V. (**E**) Liposomes for light-triggered drug delivery. Reproduced with permission from [93]. Copyright 2010, Taylor & Francis. (**F**) Liposomes for ultrasound-triggered drug delivery. The figure describes several ultrasound-mediated drug release mechanisms, including ultrasound-induced cavitation, focused ultrasound, and acoustic radiation force flow. Reproduced with permission from [94]. Copyright 2014, Elsevier B.V.

**Table 1 ijms-24-02643-t001:** Examples of traditional preparation methods of liposomes.

Methods	Preparation Steps	Characteristics of Liposomes	Advantages	Disadvantages	References
Thin-film hydration	Lipids and fat-soluble drugs are dissolved in suitable organic solvents. The solution is evaporated until the film layer is formed on the wall of the bottle, a solution with or without water-soluble drugs is added and stirred until a uniform liposome solution is formed, and the prepared liposomes can be reduced by subsequent treatment such as ultrasound and extrusion.	Preparation of MLV or LUV After sonication, unilamellar vesicles are the mainstay	Widely used and easy to operate	1. Most water-soluble compounds are washed off during swelling 2. Repeatability is not very good and not suitable for mass production 3. Low encapsulation efficiency	[4,22,23]
Reverse-phase evaporation	First, the lipids are dissolved in an organic solvent and then mixed directly with an aqueous buffer containing hydrophilic drugs. The organic solvent is evaporated under a reduced-pressure rotary evaporator and depleted to form liposomes.	LUV Reduction of size and dispersion of liposomes by extrusion		Organic solvent not easy to remove	[24]
Ethanol injection	The drug and lipid material are dissolved in ethanol, and then the lipid organic solution is injected into the high-speed stirring water phase, and the ethanol is evaporated to obtain liposomes	Preparation of unilamellar vesicles, few MLV		1. Sustained high temperatures and organic solvents may reduce the stability of drugs and lipids 2. Low encapsulation efficiency	[25]
Sonication	Treatment of prepared liposomes by water bath sonication or probe sonication	Probe sonication to prepare SUV		1. Low internal volume/encapsulation efficiency 2. High temperature easily degrades liposomes or drugs 3. Low ability to remove large molecules and metal pollution from the probe tip	[26]
Extrusion	Extrusion of prepared liposomes by liposome extruder	Preparation of unilamellar vesicles or MLV with uniform particle size by selecting polycarbonate membranes with different pore sizes	Simple process and good reproducibility	1. Laborious and time-consuming 2. Extrusion has a certain effect on the structure of liposomes	[26]
High-pressure homogenization	Homogenization of prepared liposomes by high pressure homogenizer	Mainly unilamellar vesicles	1. Good repeatability, large-scale production 2. High encapsulation efficiency and stability 3. Uniform particles		[27,28]
Freeze-thaw	Repeated freeze-thaw treatment of the prepared liposomes	Unilamellar vesicles or MLV Increasing encapsulation efficiency by repeated freezing and thawing	1. High encapsulation efficiency 2. Uniform particles	Long preparation time	[29]

**Table 2 ijms-24-02643-t002:** Examples of ligands for liposomes targeting TME.

Ligand	Targets/Applications	Reference
IL-4Rα	Suppression of tumor growth by targeting TME	[83]
Ala-Pro-Arg-Pro-Gly (APRPG)	APRPG-PEG-modified liposomes efficiently deliver SU1498 to angiogenic endothelial cells in tumors, thereby inhibiting tumor-induced angiogenesis	[84]
Anti vascular cell adhesion molecule (VCAM)-1-Fab’	VCAM-1 overexpressed in tumor vascular endothelial cells	[85]
Anti membrane type-1 ma-trix metalloproteinase (MT1-MMP) antibody	Inhibition of angiogenesis in tumor-bearing mice by targeting vascular endothelial cells and overexpressed MT1-MMP on tumor cells	[86]

**Table 3 ijms-24-02643-t003:** The most common stimuli-responsive liposomes.

Stimuli Liposomes	Stimuli	Principle	Drug	References
Enzyme-responsive liposomes	Protease, amidase, and esterase enzymes	Based on amides or esters hydrolysis by protease or esterase enzymes release loaded drugs	Oxaliplatin Vinorebine Dioscorea	[89,95,96]
pH-sensitive liposomes	pH change	pH-sensitive liposomes are typically composed of neutral lipids (usually phosphatidyl derivatives such as dioleoyl phosphatidyl ethanolamine (DOPE), dioleoyl phosphatidylcholine or N-succinyl-DOPE) and weakly acidic amphiphilic compounds such as cholesterol hemisuccinate	Curcumin Paclitaxel DNA plasmid	[87,90,97,98]
Redox-sensitive liposomes	Reactive oxygen species (ROS) peroxides, hydroxyl radicals, singlet oxygen	Depends on the redox potential difference between the intracellular reducing space and oxidizing extracellular space that occur during biological activities	Doxorubicin	[91,99]
Thermosensitive liposomes	Radiofrequency, microwave or focused ultrasound ablation therapy	Preparation of thermosensitive liposomes from phospholipids with a transition temperature of 40–45 °C, such as dipalmitoyl phosphotidyl choline, as a primary lipid with a transition temperature of 41 °C, has been employed to make these liposomes. Grafting of certain polymers may also render liposomes thermosensitive, such as poly (N-isopropyl acrylamide)	Doxorubicin Camptothecin	[92,100,101]
Light-sensitive liposomes	Ultraviolet or visible light, near-infrared	Modification of fatty acyl chains of the phospholipids with light-sensitive functional groups and the resulting phospholipids have yielded photoactivable liposomes	Doxorubicin	[55,93,102,103]
Ultrasound-sensitive liposomes	Ultrasound/high-intensity focused ultrasound	Ultrasound-responsive liposomes were fabricated by mixing nanodroplets of perfluorocarbon with PEGylated liposomes. As the pulses of US waves propagate through tissue some physical phenomena take place: cavitation, hyperthermia and acoustic streaming.	Vincristine Doxorubicin	[94,104,105]
Magnetic liposomes	Magnetic field	Metal ions and magnetic NPs (MNPs) can be combined with liposomes during synthesis or encapsulated into liposomes to prepare magnetic liposomes	Docetaxel Tegafur Doxorubicin	[106,107]

**Table 4 ijms-24-02643-t004:** Clinically used liposomes and their uses.

Clinical Products	Active Agent	Indication
Doxil^®^	Doxorubicin	Ovarian, breast cancer, Kaposi’s sarcoma
DaunoXome^®^	Daunorubicin	AIDS-related Kaposi’s sarcoma
Mepact^®^	Mifamurtide	High-grade, resectable, non-metastatic osteosarcoma
Marqibo^®^	Vincristine	Acute lymphoblastic leukaemia
Vyxeos^®^	Daunorubicin and Cytarabine	Adults with high-risk acute myeloid leukemia
Depocyt^®^	Cytarabine/Ara-C	Neoplastic meningitis
Lipusu^®^	Paclitaxel	Gastric carcinoma

**Table 5 ijms-24-02643-t005:** Influencing factors of physical and chemical stability of liposomes.

Type	Influencing Factors	Principle	Improvement Methods	References
Physical stability	Composition	Liposome properties are highly dependent on lipid composition. Liposomes with high stability can be formed by phospholipids that are not readily oxidized and hydrolyzed (e.g., hydrogenated phospholipids)	Replace unsaturated phospholipids with saturated phospholipids. The synthetic saturated lipids dimyristoyl phosphotidyl choline, dipalmitoyl phosphotidyl choline, and distearoyl phosphotidyl choline can be used to prepare liposomes that are not easily oxidized	[146]
	Particle size and distribution	Particle size and distribution uniformity of liposomes affect their stability directly	The particle size of liposomes is usually controlled in the range of 80~200 nm, and 100 nm is optimal	[147]
	Zeta potential	Appropriate zeta potential reduces aggregation and fusion of liposomes		[18]
	Phase transition temperature	When the temperature reaches the phase transition temperature, the liposome bimolecular membrane begins to be disordered from the original tight arrangement. In this case, the rigidity and film thickness of the membrane decrease, its permeability increases, and the leakage of the encapsulated contents intensifies		[148]
Chemical stability	Oxidation	Some oxidizing substances such as oxygen, and the oxidation of free radicals, easily form unsaturated fatty acid bonds in phospholipid molecules oxidative cleavage	Add antioxidants (such as vitamin E, vitamin C, flavonoids, etc.), metal chelating agents	[149]
	Hydrolysis	The phospholipid hydrolysis of liposomes is a spontaneous process, and the increase of free fatty acids in the hydrolysate reduces the pH, further promoting the hydrolysis of phosphoric acid to produce harmful substances to the human body	Preparation and preservation by freeze-drying method. Add cholesterol to slow down hydrolysis	[149,150]

## Data Availability

Not applicable.

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
