# Peer review of "Liposomes for Tumor Targeted Therapy: A Review"

_ijms, 2023, doi:10.3390/ijms24032643_

Round 1
Reviewer 1 Report
评论:
在 这份手稿,作者对作文和 脂质体的类型、制备方法和临床应用。作者 总结脂质体在组成和制备中的演变 过程,重点是脂质体从 智能脂质体的简单结构。这是一个非常令人印象深刻的 和快速增长的领域,我认为作者为一个及时的话题 《国际分子科学杂志》的广泛读者群 (IJMS)。因此,我推荐 在处理以下评论后发表本综述。
1.作者应该解释一些 缩写,例如表 2 中的 VCAM-1 和 MT1-MMP,表 3 中的 PNIPAM 和 CF127,以及 PDI在第3.2.3节中,否则部分内容不容易理解。
2. 第 4.4.5 节.:[73] 和 [79] 插入的引用不是 适当,建议用更合适的代替 引用。
3. 第4.4.5节:干细胞膜包被脂质体的描述 在本节中过于简单化。作者可以添加一些示例 间充质干细胞或补充一些其他类型的干细胞的描述 单元格以丰富内容。
4. 第 4.4.6 节.: "他等人[87]将白细胞,癌细胞和 脂质体纳米载体同时制备亮体[119]。“这句话错了。而且最后一句有些突兀,建议删减或者改 它到适当的位置。
5. 那里 是中某些单词的全名和缩写之间的矛盾 文章(例如壳聚糖、抗体、多柔比星)和作者可能 需要检查。
6. 那里 是稿件中的一些错别字(例如,聚乙二醇化和非聚乙二醇化),并且 作者可能需要检查。
Comments:
In this manuscript, the authors provide a detailed review of the composition and types of liposomes, preparation methods, and clinical applications. The authors summarize the evolution of liposomes in their composition and preparation process, with a big emphasis on liposomes evolving from simple structures to smart liposomes. This is a very impressive and fast-growing field, and I think the authors picked a timely topic for the broad readership of the Journal of International Journal of Molecular Sciences (IJMS). Therefore, I recommend the publication of this review after addressing the following comments.
1. The authors should explain some abbreviations, such as VCAM-1 and MT1-MMP in Table 2, and PNIPAM and CF127 in Table 3, and PDI in section 3.2.3., otherwise part of the content is not easy to understand.
2. Section 4.4.5.: [73] and [79] which inserted references are not appropriate and it is suggested that they be replaced by more appropriate references.
3. Section 4.4.5.: The description of stem cell membrane-coated liposomes in this section is too simplistic. The authors can add some examples of mesenchymal stem cells or supplement some descriptions of other types of stem cells to enrich the content.
4. Section 4.4.6.: “He et al. [87] fused together the membranes of leukocytes, cancer cells, and liposome nanocarriers simultaneously to prepare leutusomes [119].” This sentence is wrong. And the last sentence is somewhat abrupt, it is recommended to delete or change it to the appropriate position.
5. There is a contradiction between the full names and abbreviations of some words in the article (e.g., chitosan, antibody, Doxorubicin), and the authors may need to check.
6. There are a few typos in the manuscript (e.g., pegylated and nonpegylated), and the authors may need to check.
Author Response
Comments:
In this manuscript, the authors provide a detailed review of the composition and types of liposomes, preparation methods, and clinical applications. The authors summarize the evolution of liposomes in their composition and preparation process, with a big emphasis on liposomes evolving from simple structures to smart liposomes. This is a very impressive and fast-growing field, and I think the authors picked a timely topic for the broad readership of the Journal of International Journal of Molecular Sciences (IJMS). Therefore, I recommend the publication of this review after addressing the following comments.
Response: Many thanks for your comments. In this revision, we have tried our best to address your comments. We marked all the changes in red and point-by-point response to your comments has been listed below. Sincerely hope our revision could meet with your approval.
Point 1: The authors should explain some abbreviations, such as VCAM-1 and MT1-MMP in Table 2, and PNIPAM and CF127 in Table 3, and PDI in section 3.2.3., otherwise part of the content is not easy to understand.
Response 1: Many thanks for your comments. We have revised the manuscript according to your comment.
Point 2: Section 4.4.5.: [73] and [79] which inserted references are not appropriate and it is suggested that they be replaced by more appropriate references.
Response 2: Many thanks. We have replaced literatures in Section 4.4.5. I believe our review will be more rigorous after the corrections.
Point 3: Section 4.4.5.: The description of stem cell membrane-coated liposomes in this section is too simplistic. The authors can add some examples of mesenchymal stem cells or supplement some descriptions of other types of stem cells to enrich the content.
Response 3: Thanks for your suggestions. We have added more discussion of mesenchymal stem cell biomimetic liposomes in Section 4.4.5 in red. Hope our revision could meet with your approval.
Point 4: Section 4.4.6.: “He et al. [87] fused together the membranes of leukocytes, cancer cells, and liposome nanocarriers simultaneously to prepare leutusomes [119].” This sentence is wrong. And the last sentence is somewhat abrupt, it is recommended to delete or change it to the appropriate position.
Response 4: Thanks for this careful comment. We have revised this sentence as following: “He et al. [126] fused together the membranes of leukocytes, cancer cells, and liposome nanocarriers simultaneously to prepare leutusomes.” And we've moved the last sentence to the appropriate position in Section 4.4.5 in red.
Point 5: There is a contradiction between the full names and abbreviations of some words in the article (e.g., chitosan, antibody, Doxorubicin), and the authors may need to check.
Response 5: Thanks for your careful comment. We have revised the abbreviations in the manuscript.
Point 6: There are a few typos in the manuscript (e.g., pegylated and nonpegylated), and the authors may need to check.
Response 6: Many thanks. We have corrected the typos in the manuscript in red. I believe our review will be more rigorous after the corrections.
Reviewer 2 Report
The review article, "Liposomes for tumor targeted therapy: a review," is highly relevant, and nicely compiled. Authors have done thorough study of literature in the area. It should be published in IJMS.
Author Response
Comments:
The review article, "Liposomes for tumor targeted therapy: a review," is highly relevant, and nicely compiled. Authors have done thorough study of literature in the area. It should be published in IJMS.
Response: Many thanks for your comments. We also thank you for your time and comments.
Reviewer 3 Report
Regarding to Manuscript ID: ijms-2097332Type of manuscript: Review
Title: Liposomes for tumor targeted therapy: a review
Authors: Shile Wang, Yanyu Chen, Jiancheng Guo, Qinqin Huang * Although there are reviews with this topic, the authors documented themselves and synthesized the ideas of the topic in an original form.
Author Response
Comments:
Regarding to Manuscript ID: ijms-2097332
Type of manuscript: Review
Title: Liposomes for tumor targeted therapy: a review
Authors: Shile Wang, Yanyu Chen, Jiancheng Guo, Qinqin Huang * Although there are reviews with this topic, the authors documented themselves and synthesized the ideas of the topic in an original form.
Response: Many thanks for your comments. We also thank you for your time and comments.
Reviewer 4 Report
This review reports the progress of using liposomes for therapy of cancer. The authors focused on using liposomes for targeted drug delivery systems. The theme is of great interest, the review is divided into 6 parts, including information about classification, preparation, modification and application of Liposomes. Some minor comments:
- There are many abbreviations in the review. It will be benefit to reduce the number of abbreviations in the manuscript.
- Table 1 (preparation steps), Table 3 (principle): for comprehensive understanding, it is useful to separate in each entry with a line to avoid confuses
- The authors reported about long-circulating PEGylated liposomes. In recent years, PEGylated liposomes also used for improving mucoadhesive and mucus-penetrating properties. There are also studies about functionalizing liposomes with adhesive moieties such as maleimides, for example, to treat bladder cancer. It will be useful to add information about mucoadhesive and mucus-penetrating properties of liposomes for cancer therapy (see D.B. Kaldybekov et al. 2018, as an example)
Author Response
Comments:
This review reports the progress of using liposomes for therapy of cancer. The authors focused on using liposomes for targeted drug delivery systems. The theme is of great interest, the review is divided into 6 parts, including information about classification, preparation, modification and application of Liposomes. Some minor comments:
Response: Many thanks for your comments. We have revised the manuscript according to your comments and point-by-point response has been listed below. Sincerely hope our revision could meet with your approval.
Point 1: There are many abbreviations in the review. It will be benefit to reduce the number of abbreviations in the manuscript.
Response 1: Thank you for the introduction suggestion. We removed some unnecessary abbreviations in the review.
Point 2: Table 1 (preparation steps), Table 3 (principle): for comprehensive understanding, it is useful to separate in each entry with a line to avoid confuses
Response 2: Thanks for this careful comment. We have added a line separation to each entry to avoid confuses in Table 1 (preparation steps), Table 3 (principle).
Point 3: The authors reported about long-circulating PEGylated liposomes. In recent years, PEGylated liposomes also used for improving mucoadhesive and mucus-penetrating properties. There are also studies about functionalizing liposomes with adhesive moieties such as maleimides, for example, to treat bladder cancer. It will be useful to add information about mucoadhesive and mucus-penetrating properties of liposomes for cancer therapy (see D.B. Kaldybekov et al. 2018, as an example).
Response 3: Many thanks for your suggestion. We have added information about mucoadhesive and mucus-penetrating properties of liposomes for cancer therapy in red (4.1. Long-circulating liposomes, page 8). Hope our revision could meet with your approval.
Reviewer 5 Report
In this manuscript, Wang et al. delivered a review on the progress of liposomes for cancer therapy. The composition, types, preparation methods of liposomes, and targeting properties are well discussed, shedding light on way of their clinical applications. This review is sounding. A revised version is recommended for publication in IJMS.
1. The author should reorganize the schematic illustration in Figure 1. It’s recommended to re-draw the scheme integratively in one graph.
2. The strategies of different nanotechnology used in preparing liposomal nanoparticles should be discussed separately as a single part.
3. What’s the physical and chemical stability of liposomes like? The authors should discuss about the challenges in depth. They can list this in a table or discuss about challenges following section “5. Liposomes in clinical applications”.
4. All abbreviations through the whole manuscript should be provided with full name.
Author Response
Comments:
In this manuscript, Wang et al. delivered a review on the progress of liposomes for cancer therapy. The composition, types, preparation methods of liposomes, and targeting properties are well discussed, shedding light on way of their clinical applications. This review is sounding. A revised version is recommended for publication in IJMS.
Response: Many thanks for your constructive feedback. We marked all the changes in red and point-by-point response to your comments has been listed below. We believe that your feedback has helped us significantly improve the quality of this manuscript. Sincerely hope our revision could meet with your approval.
Point 1: The author should reorganize the schematic illustration in Figure 1. It’s recommended to re-draw the scheme integratively in one graph.
Response 1: Many thanks, according to your suggestion, we have re-drawn the Figure 1 in the manuscript (in page 2).
Point 2: The strategies of different nanotechnology used in preparing liposomal nanoparticles should be discussed separately as a single part.
Response 2: Thanks for your careful comment. In section “3. Liposomes Preparation”, we discuss in detail the conventional and microfluidic methods for preparing liposomes. The strategies of different nanotechnologies used to prepare liposome particles have been discussed as a single part in the manuscript.
Point 3: What’s the physical and chemical stability of liposomes like? The authors should discuss about the challenges in depth. They can list this in a table or discuss about challenges following section “5. Liposomes in clinical applications”.
Response 3: Thanks for your careful comment. According to your suggestion, we have added section “6. Challenges for liposomes” in the manuscript. In section 6, we discuss the physical and chemical stability of liposomes in detail, and list this in table 5. (In page 17-19)
Point 4: All abbreviations through the whole manuscript should be provided with full name.
Response 4: Many thanks for your comments. We have provided the full names of all abbreviations in the manuscript in red. I believe our review will be more rigorous after the corrections.